# MRI Banding Removal via Adversarial Training

Aaron Defazio    Tullie Murrell
Facebook AI Research,
Facebook New York

Michael P. Recht
Department of Radiology
NYU Grossman School of Medicine

## Abstract

MR images reconstructed from sub-sampled Cartesian data using deep learning techniques show a characteristic banding (sometimes described as streaking), which is particularly strong in low signal-to-noise regions of the reconstructed image. These unnatural artifacts have been identified as one of the largest obstacles preventing the clinical use of machine-learning based MRI reconstructions. In this work, we propose the use of an adversarial loss that penalizes banding structures without requiring any human annotation. Our technique greatly reduces the appearance of banding, without requiring any additional computation or post-processing at reconstruction time. Our approach is compatible with any existing reconstruction approach that uses supervised machine learning, including the current state-of-the-art. We report the results of a blind comparison against a strong baseline by a group of expert evaluators (board-certified radiologists), where our approach is ranked superior at banding removal with no statistically significant loss of detail. A reference implementation of our method is available in the supplementary material.

## 1  Introduction

The use of deep-learning approaches for accelerating MR imaging has recently shown significant promise [7, 19, 1], with learning approaches far out-performing classical penalized least squares approaches for reconstructing images from raw subsampled k-space signals. However, existing approaches produce images with some unnatural structures that prevent radiologists from accepting the images for clinical use despite the advantages the images have over classical techniques with respect to reconstruction accuracy metrics such as the structured similarity metric (SSIM).

In this work, we describe a method for removing the primary artifact produced by Cartesian deep-learning reconstruction systems: banding. This banding, as illustrated in Figure 1, is characterized by a streaking pattern exactly aligned with the phase-encoding direction (horizontal in the figure). This banding is anisotropic and non-homogenous across the image, and most visible in high-noise or low-contrast areas. It is the result of the signal subsampling process used during Cartesian accelerated MRI, whereby subsampling occurs in one spatial direction only.

At the 2019 *Medical Imaging meets NeurIPS* workshop [11], banding artifacts were shown to occur in reconstructions from each of the top 3 winning teams in the competition, despite the

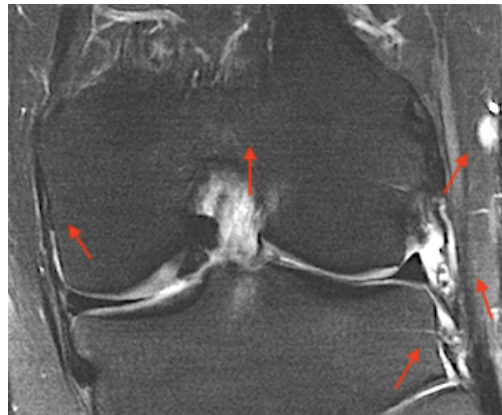

Figure 1: A deep-learning MRI reconstruction that shows significant horizontal banding artifacts

significant differences between the reconstruction approaches. These artifacts were identified as a major obstacle to the use of machine-learning based reconstructions in clinical practice.

Our paper is structured as follows: In Section 2 we formally describe the MRI reconstruction problem as it applies to current clinical MRI scanners. In Section 3 we describe how to augment standard deep-learning based MRI reconstruction methods with our orientation adversary. Section 4 describes the state-of-the-art reconstruction model that we use in our experiments. In Section 5 we describe the masking procedure we use, in Section 6 we describe the classical baseline that we compare against, and in Section 7 we detail how our model was trained. Finally in Section 8 we detail the results of a blind evaluation by radiologists.

## 2 Accelerated Parallel 2D MRI

In MR imaging, a spatial image is produced by combining measurements of the anatomy acquired in the Fourier domain, known as k-space. Let $m$ be the true greyscale spatial image. Classical approaches produce an image of the anatomy by acquiring a full cartesian grid of samples from k-space, then applying the inverse fast Fourier transform. In our notation the estimated greyscale image $\hat{m}$ is:

$$\hat{m} = \mathcal{F}^{-1}(x)$$

where $x$ is a $h \times w$ matrix of k-space measurements. In this work, we focus on 2D MRI images produced by a modern MR imaging system, which contain two additional complications: parallelization and acceleration.

### 2.1 Parallel Imaging

In a parallel MRI system, more than one receiver coil is used, resulting in a tensor of acquired k-space images. Instead of each coil imaging the entire field-of-view, each coil covers a smaller portion of the anatomy. The signal acquired by coil $i$ of $n_c$ coils is given by a Fourier transform $\mathcal{F}$:

$$x_i = \mathcal{F}(s_i \circ m) + \text{noise}, \tag{1}$$

where $s$ is a complex-valued coil-sensitivity map that is applied element-wise. The $s_i$ values can be estimated via well-known auto-calibration procedures, but this is not necessary for all deep-learning approaches.

The coil signals are commonly combined using the *root-sum-squares* (RSS) procedure to produce a spatial image. The RSS estimate at pixel $l, m$ is given by:

$$m_{\text{RSS},lm} = \sqrt{\sum_{i=1}^{n_c} |m_{i,lm}|^2},$$

where for each coil $m_i = \mathcal{F}^{-1}(x_i)$ is the individual coil image. The RSS estimate produces images with slightly (often negligibly) higher noise than more sophisticated approaches but has the advantage of robustness and simplicity.

### 2.2 Accelerated imaging

Accelerated MRI systems capture a subset of the full k-space system to reduce scan time. If parallel imaging is used, the system of linear equations given by Equation 1 is overdetermined even when a subset of the $x_i$ pixel values are known as long as the sub-sampling factor is less than the number of coils, and so a least-squares solve may be used to produce a spatial image, at the expense of an increase in noise over non-accelerated MRI [5, 18]. Two-fold subsampling is widely used in clinical practice, typically using either the SENSE [18] or GRAPPA [5] classical approaches to reconstruction. Higher acceleration factors can be achieved using regularized least-squares in the case of sparse anatomy such as vascular MRI [12], but these approaches produce poor results for general-purpose MR imaging.

### 2.3 Machine learning approaches to accelerated 2D MRI

In the machine learning approach to 2D MRI reconstruction, a training set of $n_{\text{data}}$ instances (slices) is gathered, where each instance is a k-space tensor $x^{(j)} : n_c \times h \times w$. Then standard parallel imaging

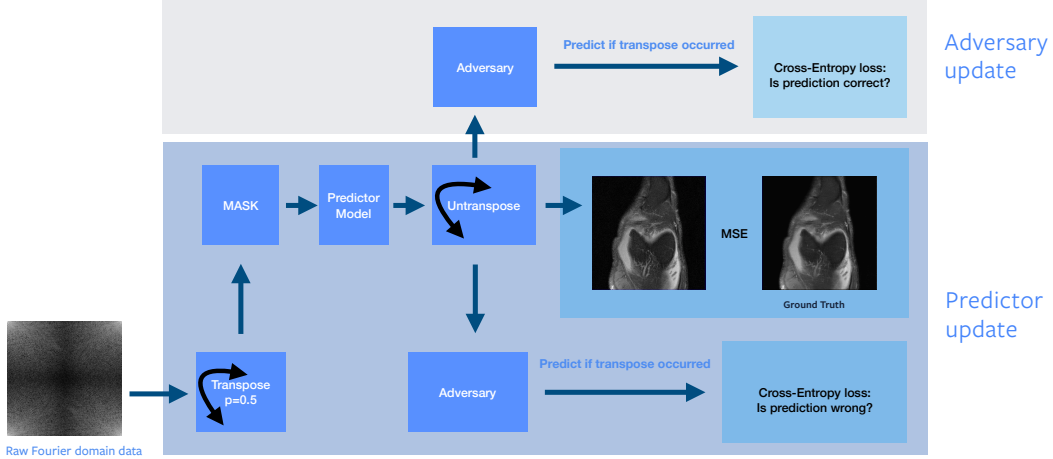

Figure 2: Orientation adversarial training

is used to produce spatial images $m^{(j)}$. A scan of a patient consists of multiple spatially consecutive slices with different scan modalities of the same anatomy, although for training purposes we treat the slices independently and sample from the total set of slices across all patients in the dataset i.i.d.

These training pairs are then used to train a black box predictor $B_\phi$, which maps from raw subsampled k-space tensors (complex valued multichannel Fourier domain images) to greyscale $h \times w$ spatial images. Given an image-space loss function $l$, the training loss for datapoint $j$ is:

$$L^{(j)}(\phi) = l\left(B_\phi\left(M\left(x^{(j)}\right)\right), m^{(j)}\right)$$

where $M$ is a masking function that zeros out a fraction of the $k$-space lines. We detail this masking function further in the Section 5. Essentially, the model $B$ is trained to produce an image as close as possible to the "ground-truth" fully-sampled parallel MRI as possible, using only a fraction of the data following the standard empirical risk minimization setup. In our experiments, we used a combination of SSIM and L1 losses with weighting 0.01 for the L1 component.

## 3   Orientation Adversary

The primary difficulty with removing banding is the lack of available annotation; direct supervised machine learning techniques can not be used. Our insight is to use image orientation during reconstruction as a self-supervised learning signal. Image banding is aligned with the direction of subsampling in the mask, in our case horizontally. This signal can be used by an adversarial training term to identify and penalize banding.

In particular, let $A_\theta$ be an adversary model, which maps from spatial images $h \times w$ to real values $[0, 1]$. Its goal is to predict if the given spatial image contains horizontal $(0)$ or vertical $(1)$ banding. We train this adversary simultaneously with the predictor model, using the same minibatch to compute stochastic gradient steps simultaneously for both, rather than in an alternating fashion.

**Predictor**   The training of our prediction model is modified as follows. Before applying $B$, a random operator is sampled using a Bernoulli variable $r^{(j)}$ with probability 0.5, either a random flip $R_1$ (transpose $h \longleftrightarrow w$) or the identity operator $R_0$. This operator is then applied before and after the application of $B$:

$$\hat{m}^{(j)} = \left(R_{r^{(j)}} \circ B_\phi \circ M \circ R_{r^{(j)}}\right) x^{(j)}.$$

The predictor's loss is then augmented with a term that encourages it to produce images that fool the adversary:

$$L_B^{(j)}(\phi) = l\left(\hat{m}^{(j)}, m^{(j)}\right) + \text{CE}\left(1 - r^{(j)}, A_\theta\left(\hat{m}^{(j)}\right)\right),$$

where CE is the binary cross-entropy. Gradients are not propagated to the adversaries parameters $\theta$ during the predictor update, but are fully propagated through the adversary from output to input then

through the predictor to its parameters $\phi$. This is accomplished by toggling requires_grad_ to false for the adversaries parameters in Pytorch during loss calculation, then toggling it back before the adversary loss calculation.

**Adversary** The adversary is trained to predict the $r$ variable directly, with the addition of a regularization term:

$$L_A^{(j)}(\theta) = \text{CE}\left(r^{(j)}, A_\theta\left(\hat{m}^{(j)}\right)\right) + \gamma \left\|\nabla_{\hat{m}^{(j)}} A_\theta(\hat{m}^{(j)})\right\|^2.$$

Gradients are not propagated through the predictor during the adversary step, using Pytorch's detach operator on $\hat{m}^{(j)}$. We found that regularization of the adversary is necessary for stable convergence. We use a simplified gradient penalty as used by [15], which is closely related to WGAN-GP penalty [6] which is widely used for Generative Adversarial Network (GAN) training.

# 4   Models

Our technique is applicable using any predictor and adversary model architecture. We detail the two models we used below. Full source code is available in the supplementary material.

**Predictor** We used the state-of-the-art predictor architecture from [20] consisting of a sequence of 12 U-Net models, interleaved with a soft-projection onto the known k-space lines. This sequence is followed by an inverse Fourier transform then a root-sum-squares operation to produce the final image. Each U-Net has 12 channels after the first convolution and has 4 pooling operations (i.e. convolutions occur at 5 resolutions when the input resolution is included). This network has only 13 million parameters, but more than 250 convolutional layers. This model is the current state-of-the-art for MRI reconstruction.

**Adversary** The shallow ResNet consists of an initial convolution to increase the channel count to 64, followed by 2 pre-activation ResNet basic blocks, 4x4 max-pooling, 2 more blocks with 128 channels, then 4x4 max-pooling followed by average pooling down to a 1x1 image, a ReLU then a linear layer. This architecture was not heavily optimized, and we expect a better architecture could give even better results. We found initially that a ResNet-50 architecture took too long to converge, and that it was necessary to include ResNet blocks that apply at the full-resolution of the image, rather than after a downsampling operation as performed in the standard ResNet architecture. Because of the small batch-sizes used for training MRI reconstruction models, we found that it was necessary to replace BatchNorm with GroupNorm [21], and we used group-size 32 as our default.

# 5   Masking

In 2D Cartesian MR imaging, the two directions in the Fourier domain are known as the Frequency and Phase encoding directions. The most common clinical practice for accelerated MRI is to only capture every second frequency-encoding line, except for a band of the lowest-frequency lines which are always included. These lines are typically stored electrically with zero values filling the unseen values. All coil images for a slice follow the same mask as the lines are captured simultaneously during acquisition. The acquisition process can rapidly acquire an entire line in the frequency direction line at once, so typically no subsampling is performed within the lines.

For the fastMRI dataset, the phase-encoding direction is always horizontal, and we use a central band of 16 low-frequency lines. These lines are only consecutive in k-space if the Fourier domain is viewed with the low-frequency lines in the center of the image, which is NOT the standard layout returned by fast-Fourier routines. The low-frequency lines serve multiple purposes [5, 18]:

- They allow estimation of the coil sensitivities through auto-calibration,
- they contain a significant fraction of the total signal energy.
- They allow easier disambiguation of aliasing produced by the mask.

Machine learning models trained using the same number of total lines but without a central region do not produce images of as high quality. The disambiguation function is important. A direct IFFT

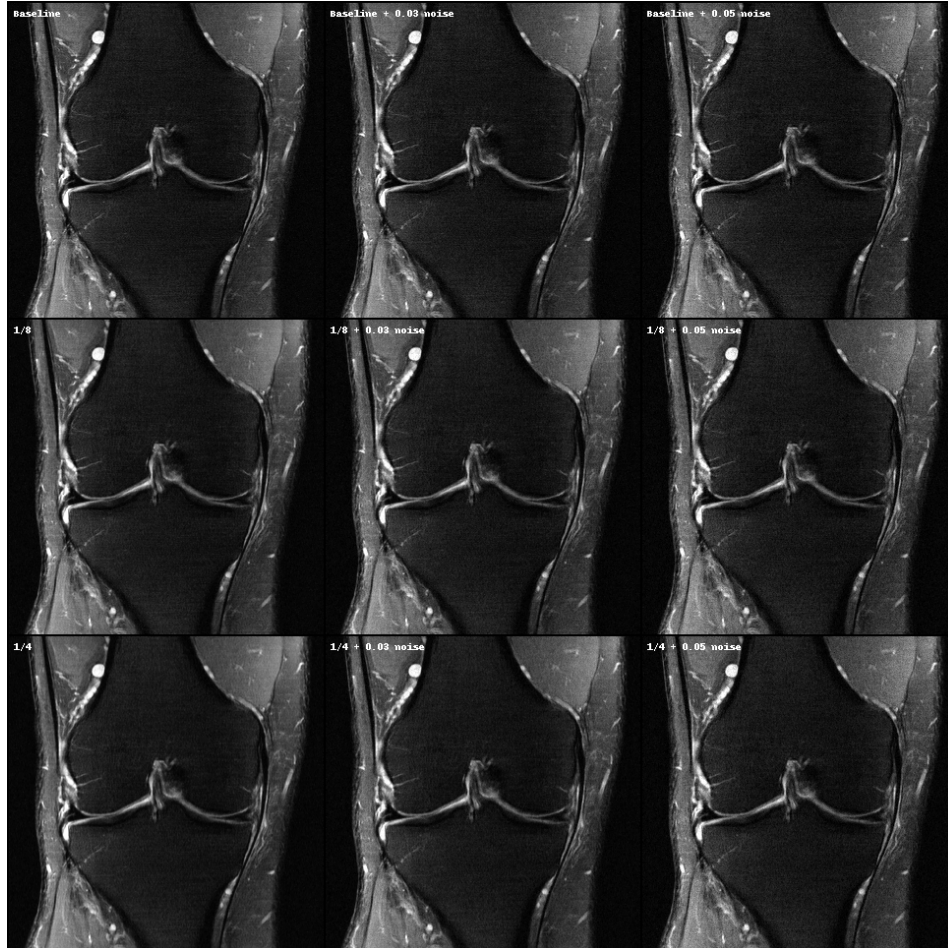

Figure 3: Baseline (top), with varying levels of blurring, against our adversarial method (bottom). Columns show different levels of adaptive noise. Images are best viewed on a high-brightness monitor at full resolution.

of a masked k-space coil image produces two images of the anatomy, with one shifted by half the image width. This is known as aliasing. Without additional k-space lines, the signal does not contain information indicating which of the two positions in the image the anatomy is in. The central low-frequency k-space lines contain information allowing a perfect reconstruction of a low-resolution version of the full coil image, which the machine learning model can use to disambiguate the two positions.

For our accelerated reconstruction we performed 4-fold acceleration by sampling every 4th frequency line, using an offset equispaced mask following [3], with the central lines as discussed above. This level of acceleration produces significant artifacts when used with classical accelerated MRI techniques, but produces very high-quality results when using deep-learning reconstructions. We also experimented with the random masks preferred by the theory of compressed sensing; however, we found they performed significantly worse. This is consistent with prior work on machine learning reconstructions [20, 3]. We attribute this to the lack of sparsity required by compressed sensing theory for the Proton Density (PD) and PD Fat Saturated (PDFS) MRI images used in our experiments.

## 6   Baselines

As the banding problem is relatively new we are not aware of any existing baseline methods for banding removal specific to MR imaging. We will instead apply a classical image processing method for minimizing the visibility of banding artifacts in images known as dithering [4, 8]. Dithering is

simply the process of adding noise in a specific way to improve the perceptual quality of an image. We found that adding noise directly to the reconstruction improved apparent sharpness, but did not by itself reduce the visibility of banding (Figure 3). However, if an anisotropic blurring filter is used before the addition of noise, banding was well-suppressed. We used the following convolutional kernel parametrized by $\alpha$:

$$K = \frac{1}{1 + 2\alpha} \left[ \begin{array}{ccc} 0 & \alpha & 0 \\ 0 & 1 & 0 \\ 0 & \alpha & 0 \end{array} \right],$$

which blurs in the vertical direction only, the opposite direction from the banding pattern. Figure 3 shows that some regions are banding free for $\alpha = 1/8$, but banding suppression improves as blurring is further increased. The addition of noise is crucial, the blurred images in the left column for $\alpha = 1/8$ and $1/4$ still show signs of banding that are much less apparent after noise is applied (middle and right column). The noise also increases perceptual sharpness, so the usage of blurring is no longer immediately apparent. The noise used here is Gaussian noise with variance equal to the median of the image in an $11 \times 11$ pixel region, multiplied by a constant.

The dithering method has the clear disadvantage of both decreasing the effective resolution of the image and decreasing the signal-to-noise ratio. This is clear when examining fine detail in Figure 3. The higher of the two noise levels shown (0.05) clearly masks banding much more significantly than a noise level of 0.03, however it was determined to be too much noise to be used clinically after consultation with a radiologist. We settled on a noise level of 0.03 as a level which is high but still a clinically relevant baseline.

Another more subtle problem arises from the inhomogeneity of the amount of banding across the image. Applying a uniform amount of dithered across the image results in some regions not receiving enough dithering, and other regions that showed no banding being unnecessarily dithered. Ideally, the dithering would be applied adaptively across the image, in proportion to the amount of banding; however, this requires a model or method that outputs the "banding amount" in a region, which is a non-trivial problem. Our proposed adversarial method implicitly learns such a model, as part of the discriminator term in its loss.

Since dithering can improve the perceptual image quality as assessed by a viewer while at the same time reducing signal-to-noise, resolution, and measures such as MSE and SSIM compared to the ground truth image, its use must be carefully considered. We determined that blurring up to $\alpha = 1/8$ may be of use clinically but that $\alpha = 1/4$ losses too much fine detail.

## 7 Training

We trained our models on the knee scans from the fastMRI dataset [22] using the approach detailed above. This is the first large-scale dataset released of raw full-sampled k-space. For the evaluation, we trained and evaluated on scans from 1.5 Tesla machines only as banding is less visible in scans from 3 Tesla machines making evaluation more difficult, although our approach works equally well for 3T scans. Training consisted of 100 pre-training epochs, where the adversarial term is not used (but the flip operator is included), using ADAM with learning rate 0.0003, momentum 0.9 and batch-size 8 (1 per GPU on an 8-gpu system) and no weight decay. The pre-training was followed by 60 epochs of training including the adversarial term, with learning rate 0.0001, and adversary gradient regularization strength $\gamma = 0.1$. We trained with a factor of 4 acceleration, using 16 central k-space lines, using the train/test splits distributed as part of the fastMRI dataset.

## 8 Evaluation by Radiologists

We set up our reader study as a three-fold comparison against the adversarial method, the non-adversarial method, and the dithering baseline. The non-adversarial method, which uses an identical setup to the adversarial approach excepting the adversarial terms in the loss, will be denoted the "standard" approach. Each of the six board-certified radiologists in our study were given a set of 20 of 40 volumes from the validation set (Each volume is approximately 25 slice images), so that each volume of the 40 was evaluated three times independently.

Each radiologist was asked to rank the three methods in terms of the degree of banding and separately in terms of the retention of fine detail (the questionnaire given to each radiologist is in the Supple-

| p-values | Standard | Dithering | Average rank (higher is better) |
|---|---|---|---|
| Adversary | $1.09 \times 10^{-11}$ | $2.18 \times 10^{-11}$ | **2.83** |
| Dithering | 0.028 | - | 1.74 |
| Standard | - | - | 1.43 |

Table 1: `Banding Removal` – results of the two-sided pairwise comparison with Bonferroni correction for the following question proposed to board-certified radiologists: "**Rank the 3 methods in terms of the amount of visible banding**".
Our proposed method is ranked as better than the two baselines with very high statistical significance.

| p-values | Standard | Dithering | Average rank (higher is better) |
|---|---|---|---|
| Adversary | 2.61 | $8.82 \times 10^{-4}$ | **2.18** |
| Dithering | $3.25 \times 10^{-6}$ | - | 1.5 |
| Standard | - | - | **2.32** |

Table 2: `Detail retention` – results of the two-sided pairwise comparison with Bonferroni correction for the following question proposed to board-certified radiologists: "**Rank the 3 methods in terms of the retention of fine anatomical detail in comparison to the ground truth**". Our proposed method is not statistically significantly different in terms of detail retention from the standard baseline, and highly statistically significantly better than the dithering approach. P-values range from 0 to 3 due to the Bonferroni correction.

| Adversary | Standard | Dithering |
|---|---|---|
| **0.725** (0.62-0.82) | 0.967 (0.92-1) | 0.975 (0.90-0.99) |

Table 3: `Presence of banding` – two-sided 95% binomial confidence intervals with Bonferroni correction for the question proposed to board-certified radiologists "**Is any banding present (yes, no)**". Our approach completely removes all traces of banding 27.5% of the time. The standard and dithering approaches are rarely reported as having no banding by radiologists (<4% of the time).

mentary Material) for the volumes provided to them. Equal adaptive noise as detailed in Section 6 was used on all three to avoid the known bias of human evaluators to assess noisy images as having higher detail levels.

They were given access to a ground-truth corresponding to the non-accelerated (fully sampled) images. For each volume, the methods were assigned designations "A", "B", "C", randomized at the volume level, to ensure that the evaluators were blind to the reconstruction method used. A two-sided paired sign-test with Bonferroni multiple comparison correction to the p-values was used. The tests were performed using the average rank from the 3 radiologist who ranked each volume. Volume evaluations are considered independent for the purposes of the test, and the pairing is at the within-volume level. The statistical analysis was chosen at the experimental design stage to avoid statistical fishing. The raw experimental results and R script used for our analysis are given in the Supplementary Material.

## 8.1 Evaluation Results

The radiologists ranked our adversarial approach as better than the standard and dithering approaches with an average rank of 2.83 out of a possible 3. This result is statistically significantly better than either alternative with p-values $1.09 \times 10^{-11}$ and $2.18 \times 10^{-11}$ respectively, and the adversarial approach was ranked as the best or tied for best in 85.8% of 120 total evaluations (95% CI: 0.78-0.91). The dithering approach is also statistically significantly better than the standard approach.

We also asked radiologists if banding was present (in any form) in the reconstructions in each case. This evaluation is highly subjective, as "banding" is hard to define in a precise enough way to ensure consistency between evaluators. Considering each radiologist's evaluation independently, on average banding is still reported to be present in 72.5% (**95%** CI: 0.62-0.82) of cases even with the adversarial learning penalty. The radiologists were not consistent in their rankings; the overall percentages reported by the six radiologists were 20%, 75%, 75%, 80%, 85%, and 100% for the adversarial reconstructions. In contrast, for the baseline and dithered reconstructions, only one radiologist reported less than 100% presence of banding for each method (80% and 85% presence respectively, from different radiologists).

Ground Truth        Standard        Orientation Adversary

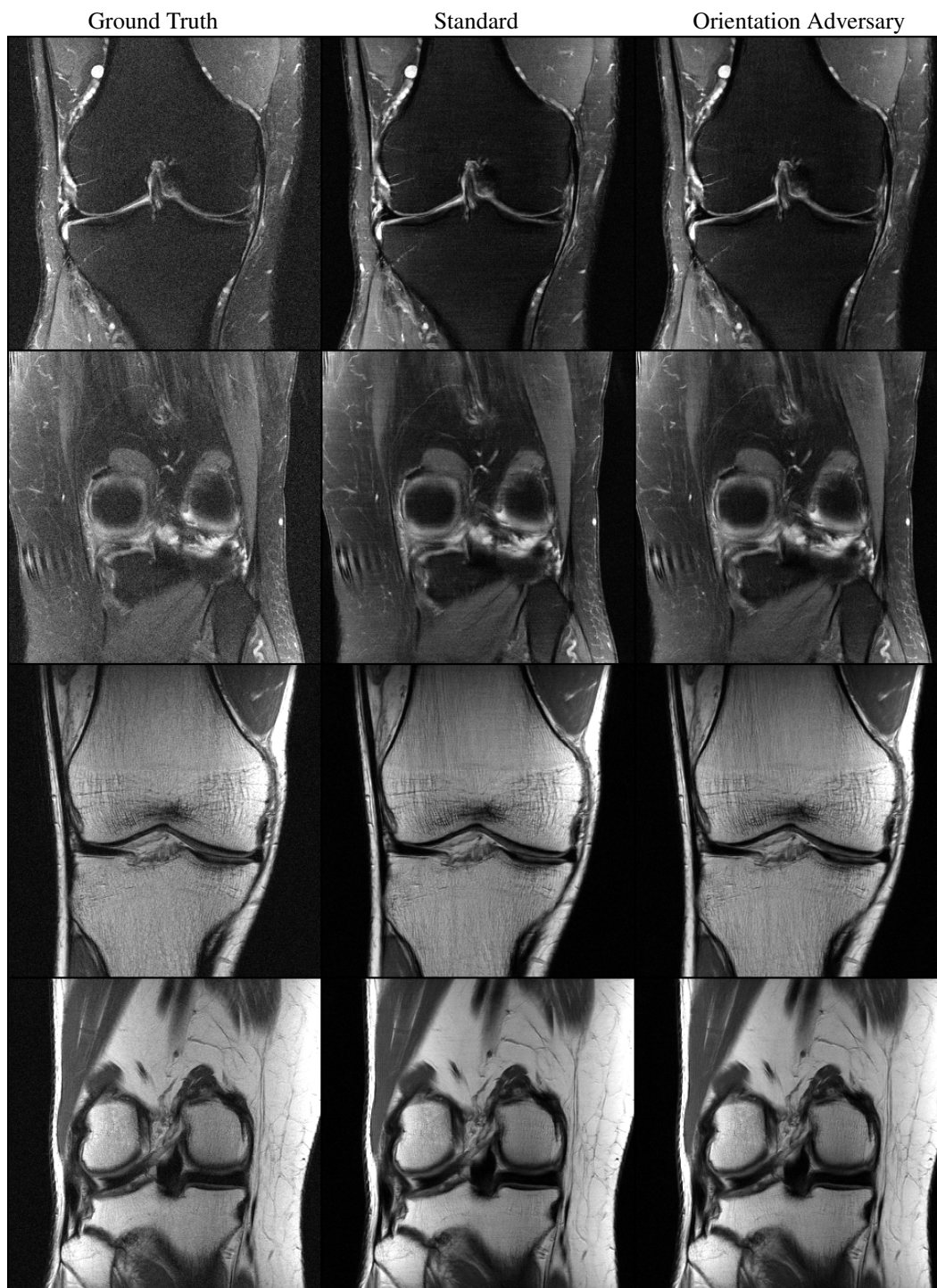

Figure 4: A comparison of ground-truth images against a standard accelerated deep-learning recon-struction and the proposed orientation adversary. Two fat-suppressed and two non-fat suppressed images are shown, chosen at random from the validation set. Images are best viewed on a high-brightness monitor.

# 9   Related Work

Machine-learning based reconstruction of MRI images is an ongoing research direction and not currently in clinical use, so very little discussion of the downsides to current techniques exists in the literature. We are not are of any work on banding removal of accelerated MRI images; however, other forms of banding introduced by balanced steady state free precession (bSSFP) sequences have been investigated, using both physics-based [2] and supervised machine learning [10] approaches. These techniques are not directly applicable to the removal of acceleration banding artifacts.

Adversarial learning has been applied to medical imaging in a number of recent works, for instance for segmentation [16], Magnetic Resonance Angiography (MRA) image generation [17], super-resolution [13], and generation of anonymized datasets [9]. Adversarial training has also been applied directly for MRI reconstruction [14]. We believe caution should be used when applying adversarial penalties to directly encourage reconstructions to resemble non-accelerated ground-truth images, they are prone to introducing phantom anatomical detail. Our use of an adversarial penalty to detect orientation does not suffer from this problem as it only ever takes reconstructed images as input rather than ground-truth images.

## Conclusion

In this work, we have presented an effective technique for producing machine learning models for accelerated MRI that minimize visible banding artifacts. Our technique is **not** specifically tied to any reconstruction approach, it may be be applied on top of any MRI reconstruction technique that uses supervised machine learning.

## Acknowledgements

This work was made possible through close collaboration with the fastMRI team at Facebook AI research, including Nafissa Yakubova, Anuroop Sriram, Jure Zbontar, Larry Zitnick, Mark Tygert and Suvrat Bhooshan, and our fastMRI project members at NYU Langone Health [22], with special thanks to Florian Knoll, Matthew Muckley and Daniel Sodickson. In addition to each author's institution, this work was supported by National Institutes of Health grants R01 EB024532, R21 EB027241, and P41 EB017183.

## Broader Impact Statement

The use of machine learning in medical imaging raises a number of ethical and societal considerations. The benefits to the use of our method are clear; images produced using our method have fewer image artifacts than standard reconstructions. Our method, used in combination with current machine-learning based reconstruction methods, has the potential to significantly reduce time spent in MRI scanners, and the associated cost of scanning. This benefits both patients and medical professionals.

In theory, the removal of the banding artifacts may lead to the occlusion of fine anatomical detail that would otherwise be present and useful for diagnosis. Our evaluations by radiologists are strong evidence that the method does not remove detail, but we believe a larger study directly aimed at determining differences in diagnosis would be required to establish this. It's not a-priori certain that any method of banding removal will reduce detail, since the ground truth does not have banding, the use of a banding removal method is just a use of prior knowledge in the Bayesian sense.

Machine learning approaches to MRI reconstruction are potentially prone to biases in the training data. If anatomy outside of the norms of the training data is encountered, the reconstruction may not be accurate. When using our proposed approach, the same considerations are necessary as for any application of machine learning in medical imaging. At a minimum the model must be tested for robustness to outlier examples.

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
