[Reviews · NeurIPS 2020]

Review 1

Summary and Contributions: The paper describes an approach to undersampled MRI reconstruction using deep learning that is designed specifically to avoid a particular banding artefact that commonly arises in current learning-based approaches to this problem. The approach uses an adversarial strategy to force the reconstruction network to produce images in which an adversary cannot determine the orientation of the banding artefact. Experiments use qualitative assessment by radiologists on the presence of the artefact and overall image detail in images reconstructed with the new approach, with a standard approach, and using a more traditional artifact removal approach through blurring. Results are somewhat equivocal on level of detail retained, but do show that the artifact is effectively reduced.

Strengths: The paper addresses a real problem in a high impact application area – MRI is widely used in medical imaging and would benefit substantially from effective machine-learning approaches to reconstruction from sparsely sampled data. The approach is innovative within its application domain and draws on modern ideas in machine learning that have not been applied in this way before. The experiment design is well focused on the precise aims of the work.

Weaknesses: I’m left unconvinced this is really the right way to deal with a persistent artifact. The broad strategy is to force the reconstruction to find a way to make the artifact invisible, whereas I believe users would be more comfortable with an approach that really understands why the artifact arises and mitigates it in a more informed way. Further to the point above, how do we know the adversarial training doesn’t introduce some other kinds of more subtle artifact while masking the streaking artifact? While the experiments are focused on the key aims, as noted above, they neglect wider impact of the proposed reconstruction. Are other artifacts introduced (question not asked)? Would radiologists in reality be more comfortable with seeing the banding artifact than seeing the output of an alternative processing step that removes it in a way that is not fully understood? In this sense the dithering approach may be preferable even if not quite so effective at obscuring the artifact. Results aren’t hugely compelling. While the proposed method certainly reduces perception of the artifact (table 1), it retains less detail than the standard approach (table 2). I do respect the authors honesty about this, but it certainly suggests more work is needed here, and I wonder if this paper is more suitable to a specific MRI conference or journal rather than being ready for exposure to a more generalist audience like NeurIPS. More generally, while this paper will be of clear interest to the MRI community, I’m less sure it is of general interest to the NeurIPS community. It is an application of largely existing machine learning ideas tailored to a very specific albeit important problem in MRI. We learn very little about machine learning in general and no attempt is made in the paper to explain wider impact or potential of the ideas in other applications or areas of machine learning.

Correctness: I did not pick up any substantive mistakes.

Clarity: It’s OK. The ideas come across fine, although there are some niggles: - It is hard for the uninitiated to appreciate the artifact in figure 1. Would be better to see the image side by side with a traditional reconstruction where the artifact isn’t present or somehow to highlight what we are looking at. - The persistent use of “MRI imaging” (expands to “magnetic resonance imaging imaging”) is jarring. - The last paragraph of the introduction misses out some sections, which is weird. - The mathematical notation is a bit of a mess. m is undefined in section 2 and seems to be used both to represent the “true” image function that the MR image estimates and an index into that function (see second equation in section 2.1).

Relation to Prior Work: The referencing of deep learning for MRI reconstruction is sparse, but I guess fine to put the ideas of the work in context.

Reproducibility: Yes

Additional Feedback: Following rebuttal and reviewer discussion I appreciate that this has been an emerging topic in NeurIPS workshops so of some interest to the community. Still not sure the interest spreads beyond a small subset, but nevertheless, I've increased my score to just above acceptance and reduced my confidence slightly.


Review 2

Summary and Contributions: The present paper proposes to tackle a common issue faced by machine-learning based MRI reconstruction algorithms (when using entire line subsampling): the apparition of banding artefacts. An adversarial based reconstruction method is proposed. k-space data is randomly flipped before undergoing an undersampling process, and then reconstructed by (any) ML reconstruction framework. An adversarial network is added and trained to distinguish between horizontally and vertically flawed images. By trying to fool the adversarial network (integrated in the predictor loss), the reconstruction network learns to minimise bandings. The method is assessed with automated score (SSIM) and by visual inspection by expert radiologists, as was performed and suggested during the 2049 Medical Imaging meets NeurIPS workshop challenge. Three different methods (proposed / Standard / Dithering ) are used to assess the performance of the proposed approach.

Strengths: -The paper is very well written, the problem of accelerated MRI reconstruction is thoroughly introduced making the paper accessible to the NeurIPS community. The proposed methodology is defined in details and the theoretical approach is well justified. - The authors propose an original and creative way of removing banding artifacts, by introducing a random flip before undersampling. - Banding effect is of interest to the community as all top entries in the 2019 challenge were exhibiting such artefacts. To my knowledge, no suitable solution has yet been proposed. - Performances of the proposed method are significantly better for banding removal than the other approach tested without decreases the detail retention. - The adversarial training method is compatible with any existing machine-learning reconstruction algorithm.

Weaknesses: - The dithering method used for comparison purpose has been optimized using the feedbacks of a single radiologist. The authors stated that adding extra noise is more effective for banding artefact removal but could lead to artefact details suppression. It would have been interesting to compare the proposed approach with the dithering method with two different levels of noise in order to back up this claim. - As stated by the authors, the visual inspection of the images require high-brightness monitors. Using standard monitors it is quite complicated to spot differences in the different approaches, and without radiologist expertise it is impossible to rate the different techniques. - The authors used only the images acquired on a 1.5T scanner, as images acquired on a 3T scanner seem less prone to banding artefacts according to the authors. Does that not suggest that the problem is not so significant (given the high prevalence of 3T clinical scanners? How do the authors explain this difference? Does it come from the higher SNR on 3T scanner? - The authors only applied their banding artefact removal technique on a single reconstruction model (state-of-the-art method)? The proposed method is applicable to any machine learning technique, and it would have been interesting to see how it would have performed on at least another machine learning technique. - The authors are stating that these banding artefacts are one of the most important issues preventing the adoption of machine-learning recon algorithms in clinical practice. I have the feeling that clinicians are much more worried by the robustness and stability of such ML recon techniques for pathological cases, or their capacity to render small low contrast pathological features (such as subchordal osteophyte or meniscal tear), as discussed in the 2019 challenge paper. Examples on the 2019 challenge paper seem to indicate that these small features disappear on all the top entries of accelerated reconstructed images. Wouldn’t it more relevant for the clinicians to evaluate these recon schemes not visually assessing the image quality but by comparing the clinical diagnosis by multiple expert radiologists? Demonstrating that radiologist are able to spot pathologies on accelerated reconstructed images as accurately as on standard images would have a significant impact on the clinical community? Does the current challenge dataset contain enough pathological cases and also contain this diagnosis information in order to perform such an evaluation? - By closely inspecting the images, it seems that the proposed method is creating also vertical banding artefacts which when combined to horizontal banding artefacts create a less structured noise (close to salt and pepper noise)? Is it something that the authors have also noticed or are these vertical bands only appearing in my mind? - The fully raw sampled data is required in order to train the proposed technique, which is also required for supervised based ML recon techniques? How would the authors deal with a dataset of under-sampled raw data (for applications where fully sampled is not feasible, maybe cardiac data for instance)?

Correctness: The authors have followed the empirical methodology suggested during the 2019 challenge, and such methodology is sound.

Clarity: This paper is very well written and easily comprehensible. Each step of the process is well described and justified. I particularly appreciated the introduction on accelerated reconstruction principles, making the manuscript accessible to the NeurIPS community.

Relation to Prior Work: The originality of the proposed manuscript is clearly discussed, and the ML reconstruction used in this paper is well described and clearly referred.

Reproducibility: Yes

Additional Feedback: - Wouldn’t it be possible to add a term in the preditor loss that penalise high horizontal gradients? - The authors assume that each slices from the stack of data is iid? Does this strong assumption induce some too string morphological prior on the reconstruction? - Would it not be possible to use adjacent slices with a different subsampling strategy in order to reconstruct multiple slices simultaneously? The locations of the branding artefacts are dependent on the position of the lines being sampled or not, by using different lines on adjacent slices the artefacts would located elsewhere in the image and it would be able to “smooth” in the reconstruction by using the information from the adjacent slice? - The authors should specify how many images were used for training and testing, and also if they (as I assume) used the partition of the dataset (train, validation and test) as the 2019 challenge. - Images are impossible to be analysed by non-experts, indicating the area of importance and pathological features would help the lay readers to interpret them.


Review 3

Summary and Contributions: The authors propose a method to remove banding artifacts from reconstructed MRI images. The authors propose a self-supervised, adversarial approach that learns to identify and penalize this king of artifacts. The authors validate the contribution with a group of expert radiologists. The technical contribution is rather simple: time image is flipped with probability 0.5, and the adversary must learn to predict whether the flip happened, and the predictor is encouraged to produce images that don't exhibit banding in order to fool the adversary.

Strengths: The proposed methodology to overcome the problem of banding seems sound, relying on standard techniques of adversarial techniques. The technical contribution is rather straightforward, but seem to provide good results. I especially like that the fact that the performance was evaluated by clinicians, which is far too rare in the latest deep-learning based approaches for medical imaging reconstructions. The problem has been rarely considered, as most work focused mainly on reconstruction as a whole, and the authors introduce a sound baseline as a comparison. The overall work could potentially be impactful in the MRI community.

Weaknesses: POST DISCUSSION: After the authors' rebuttal and discussion with the other reviewers, I was convinced to raise my score to an accept, especially because this banding artefact was something highlighted in the NeurIPS medical imaging workshop, and is thus relevant for the participants. However, I still think that trying to show the relevance of this technique beyond the medical imaging community could make the contribution overall stronger. ----- While the contribution is certainly valuable, the paper mostly clear and the methodology well executed, I think that NeurIPS is not an adapted venue for this work. There are several lines of concern regarding this paper. The first one is, in my opinion, that banding removal is *not* very relevant beyond the very specific application of MRI, and I think that it is critical that the authors should motivate why such an approach could be relevant to the broader ML community. However, I have the feeling that some readers of the paper might not even be able to see what banding artifacts actually are, as to me, it seems like a subtle kind of artefact exhibited by reconstructed MR images. For instance, it might not be clear on Figure 1 what these "significant horizontal banding artifacts" are, and I would encourage the authors to clearly highlight it, by putting a reference image without banding and possibly an error map. Secondly, I think that many references are missing. The authors do not cite early CS work in MRI (e.g. [1] below), nor references for parallel imaging (e.g. [2] below). No references are cited on the uses of the low-frequency lines at l. 147-149 (one could cite [2,3] below). Similarly, while dithering is a classical image processing technique, a reference to a book or classical paper could be useful. Again, such references would be of great help to the non-expert reader. I would have also enjoyed seeing empirical validation on more than a single model, to show the generalizability of the technique beyond ref. [15]. In addition to this, I think that a quantitative evaluation of the method in terms of standard metrics such as SSIM or MSE could provide an insightful view to non-experts readers. In addition, the authors mentioned to have optimized their method on a mix of SSIM and L1 losses, and experiments using a standard L2 training could also have been useful to show the robustness of the banding removal. [1] Lustig, Michael, David Donoho, and John M. Pauly. "Sparse MRI: The application of compressed sensing for rapid MR imaging." *Magnetic Resonance in Medicine: An Official Journal of the International Society for Magnetic Resonance in Medicine* 58.6 (2007): 1182-1195. [2] Pruessmann, Klaas P., et al. "SENSE: sensitivity encoding for fast MRI." *Magnetic Resonance in Medicine: An Official Journal of the International Society for Magnetic Resonance in Medicine* 42.5 (1999): 952-962. [3] Griswold, Mark A., et al. "Generalized autocalibrating partially parallel acquisitions (GRAPPA)." *Magnetic Resonance in Medicine: An Official Journal of the International Society for Magnetic Resonance in Medicine* 47.6 (2002): 1202-1210.

Correctness: I don't have any major concern with methodology and claims of the paper. I think that the Bonferroni multiple comparison correction to the p-values is a good practice for the statistical evaluation, I like the fact that they used it. I also like the precision that the statistical analysis was chosen at the experiment design stage, it is a good practice. Overall, the both the method and empirical methodology seem solid.

Clarity: I think that the paper is highly saturated with MRI-specific concepts, that might be quite inaccessible to non-experts. Here are a couple of examples of this: - In the introduction, the authors mention "raw subsampled k-space signals", but it might not be obvious to the reader that "raw" refers to complex images and that k-space is the Fourier domain (defined in section 2.1). - Then, the authors define banding as being characterized by "streaking pattern exactly aligned with the phase-encoding direction", but the phase encoding direction is only defined in passing at the beginning of Section 5. If the authors are convinced that NeurIPS is the correct venue for their work, I would strongly encourage them to make their paper more accessible to non-experts. I think that the CS-MRI concepts are introduced a bit quickly, and that it might help the reader to have a proper mathematical description of the problem, with subsampling being defined as a dimensionality reducing operation in the Fourier domain. In section 8, could the authors clarify what is meant by the Standard approach? I guess that it is simply performing reconstruction without any adversarial loss and postprocessing. Is it correct?

Relation to Prior Work: As there is not much prior art in banding removal, the authors resort to dithering, a classical image processing technique used for minimizing visual artefacts. However, as I said before, I would have like to see references to books/papers discussing this technique. In addition, as I also mentioned above, I think that the paper does not references the classical works in MRI upon which it builds (CS-MRI, parallel MRI).

Reproducibility: Yes

Additional Feedback: In relation to NeurIPS not being the right venue for the paper, I would also like to add that I don't think that ML researchers would be the best audience to properly appreciate the care taken in the modeling of the problem. For instance, working on multi-coil data instead of the the simpler setting single-coil data is something valuable that might be under-appreciated. **Various remarks: - The caption in Figure 3 is too small. - At l. 162-163, the authors said that they "also experimented with the random masks preferred by the theory of compressed sensing; however, [they] found they performed significantly worse." Could they be more specific about their claim? - l. 164. Proton density (PD) and PD Fat Saturated (PDFS) are used without the acronym being defined. **Note on the score: I think that the paper is well written and the experiments well executed. If the authors were to make their contribution accessible to the non-expert as well as motivate applicability of their method beyond MRI, I would be willing to substantially raise my score.


Review 4

Summary and Contributions: Deep learning based approaches to MRI image reconstruction cause banding/streaking that affects the quality of the reconstructed image. The authors propose methods that reduce the appearance of these artifacts.

Strengths: The authors are clear in stating the problem they seek to solve, and their proposed method is shown to alleviate the problem of banding artifacts in MRI reconstructions. The paper provides a straightforward comparison to other methods, and is a good starting point for other researchers to benchmark their methods in the future. MRI reconstruction using deep learning is an important field of research that should receive attention due to its potential clinical impact.

Weaknesses: 1. Motivation: The authors acknowledge that the banding problem is new, and so more work needs to be done in this paper to establish motivation. While medical image reconstruction ideally does not contain artifacts, it's not clear why banding artifacts in MRI are clinically prohibitive. (Other medical imaging reconstructions contain artifacts, but can still be used in clinics). This paper is not claiming to solve the artifacts problem, so does it alleviate it enough to make a clinically relevant difference? The radiologists surveyed in this paper are ranking images for artifacts, which is not an actual task they would perform in clinics. Measuring radiologists' performance on clinical decisions (eg. detection) on images subject to different removal methods would help answer this question. In lieu of this, the paper could be still be improved by referencing similar medical imaging modalities where artifacts have been proven to hinder radiologist's performance, since the same pattern recognition processes are involved. (Alternatively, providing evidence that artifacts present a regulatory barrier to using deep learning for reconstruction would also suffice). 2. Baseline comparisons A. While banding is a new problem within the domain of MRI reconstruction, removing artifacts from medical images is a relatively well-explored domain. Why can't general techniques used to remove artifacts in reconstructed MRIs be applied to banding? B. Papers such as Bau, et al have explored why generated artifacts occur, and propose potential approaches that involve removing "artifact" units. The comparison section would be stronger if it included attempts to resolve the banding problem through changes in the predictor model's U-Net architecture itself. 3. Radiologist evaluation There should be a control group of non-deep learning reconstructions that do not contain banding artifacts.

Correctness: The correct statistical tests are used to evaluate the experiment. The use of six board-certified radiologists adds scientific rigor to the visual testing. The model description is adequate and the training procedure is described well enough to reproduce the model in a different setting.

Clarity: The paper tells a clear story by outlining what it sets out to solve and how it seeks to do so. As someone who is not well-versed in MRI reconstruction, but is familiar with medical imaging, I found it hard to understand some of the jargon related to MRI reconstruction, especially in section 5 (Masking).

Relation to Prior Work: The paper is a first attempt at removing banding artifacts from deep learning approaches to MRI reconstruction, so it does not have direct prior work. However, it still falls within the realm of more general attempts of using adversarial training to remove artifacts in medical images, and needs to more clearly explain why a tailor-made solution is needed here.

Reproducibility: Yes

Additional Feedback: I believe this paper is a marginally above the acceptance threshold. For it to be considered a higher score, more work needs to be done in the clinical significance of removing banding artifacts. A closer analysis as to why banding artifacts occur in CNNs and an attempt to remove them by improving the underlying architecture would make this a stronger deep learning paper. My concerns regarding the clinical significance of this paper remain, still maintain an overall score of 6.

[Author Response · NeurIPS 2020]

**Relevance** I would first like to start by commenting on the relevance of this work to the NeurIPS community, as this was a concern raised by two reviewers. NeurIPS has a long history of accepting 'application' papers, and I believe that our paper falls well within the category 'Applications -> Health' as listed in the call-for-papers. The workshop 'MEDICAL IMAGING MEETS NEURIPS' was held in 2017,2018 and 2019; and was well attended. We decided to submit this work to NeurIPS because it directly addresses the banding issue that was first identified in the 2019 workshop, and we believe that many researchers working on the fastMRI dataset that we use will be attending the conference (virtually) this year as well, as a second fastMRI competition has been announced as part of the 2020 workshop.

I would like to urge the reviewers to discuss this relevance issue with each other, and contact the area chair for guidance if necessary. This is a difficult issue to decide, and I hope you can take into account that the NeurIPS community is large and diverse.

**Response to reviewer 1** Thank you for the detailed feedback, it is much appreciated. You're absolute right that it is very difficult to access the quality of the images generated by a reconstruction method when there are conflicting concerns. A method that produces visually pleasing results but introduces additional artifacts is of little use in practice. The focus of our approach from the beginning was to avoid introducing artifacts, which is why we took the approach of an orientation adversary, rather a more traditional GAN style adversary. Our early experiments with a GAN approach showed that it introduced false detail into the reconstruction.

We believe our approach is actually motivated by the process producing the artifacts. As we conjecture, the artifacts are due to the orientation of the sampling process. This is why a loss that penalizes orientation-aligned artifacts is sensible.

We did not ask the panel of radiologists to directly assess if new artifacts had been introduced, and in retrospect I believe you are right, it would strengthen our work if we had. I can say however that we have not seen any evidence that our method introduces new artifacts, and we have looked at a large number of reconstructions from the model. We have also regularly consulted with a radiologist (separate from the panel) who did not identify any issues. We would be more than happy to add a discussion of this to the paper to address your concerns.

Also, we would like to note that Table 1 doesn't shown that our method retains less detail. The difference in detail scores is actually within the statistic margin of error, as shown by the p-value listed in the table. Note the p-value is above 1 due to the Bonferroni correction, but it would still be much larger than 0.05 even without the correction.

**Response to reviewer 2** We have not been able to answer all your questions due to lack of space. In regards to the use of 1.5T scans for the experiments, we do believe it is the signal-to-noise ratio that primarily contributes to the greater degree of banding compared to 3T scans. Our focus on 1.5T scans is motivated by clinical practice and the direction the field is taking as a whole. They are still very common due to the lower cost, and they must be used in cases where the patient has implants that have not been certified as safe for 3T scanning. Using machine-learning to enable even lower field scanners is currently an active research area, including portable scanners which are at the prototype stage.

You are absolutely correct that a larger scale study of the ability of radiologists to still detect pathologies in the reconstructed images should be considered the gold-standard. However, this sort of comparison is extremely expensive as it requires a much larger amount of radiologist time to achieve statistical significance, it's just not possible for us to do such a comparison.

Unfortunately, our method can not be applied in cases where the data is undersampled. We have spent a lot of time trying to find a solution and we have not been able to come up with any method that works without full data for training.

**Response to reviewer 3** We hope we have addressed your concerns with relevance above, and we hope you can reconsider your rating in light of our comments. We agree that we need to provide a more detailed introduction to the terms and concepts used so that our paper is more approachable. We can fully commit to improving the readability of our work for the camera ready.

We will also add a discussion of how our approach builds upon classical compressed sensing. As you note, there is some surprising differences between deep-learning approaches and classical approaches with regards to the masks. Equispaced masks out perform random masks for deep-learning reconstructions (SSIM 0.926 vs 0.915 for our model). The recently released brain portion of the fastMRI dataset moved to equispaced masks from the random masks used in the initial knee release for this reason.

**Response to reviewer 4** Please see our response to reviewer 2. Our focus on reducing this particular artifact was initiated by our regular consultation with practicing radiologists, they have told us that it is a very major concern they have with the reconstructions, second only to the retention of anatomical detail and pathologies.

[Meta-Review · NeurIPS 2020]

Initially the four reviewers have mixed opinions, mainly concerning the paper's relevance to the NeurIPS community. Hwoever, the rebuttal changes their opinions to positive as the topic of MRI Banding Removal or MRI reconstruction in general fits nicely into the workshops such as "MEDICAL IMAGING MEETS NEURIPS". I have no reason to go against that and hence recommend acceptance.